# Evaluating a Soil Amendment for Cadmium Mitigation and Enhanced Nutritional Quality in Faba Bean Genotypes: Implications for Food Safety

**DOI:** 10.3390/plants14010141

**Published:** 2025-01-06

**Authors:** Liping Cheng, Jiapan Lian, Xin Wang, Mehr Ahmed Mujtaba Munir, Xiwei Huang, Zhenli He, Chengjian Xu, Wenbin Tong, Xiaoe Yang

**Affiliations:** 1Ministry of Education (MOE) Key Laboratory of Environmental Remediation and Ecosystem Health, College of Environmental and Resources Sciences, Zhejiang University, Hangzhou 310058, China; lpcheng@zju.edu.cn (L.C.); ljiapan@zju.edu.cn (J.L.); 22214128@zju.edu.cn (X.W.); muju212@mail.ustc.edu.cn (M.A.M.M.); 22114147@zju.edu.cn (X.H.); 2State Key Laboratory for Conservation and Utilization of Subtropical Agri-Biological Resources, Guangxi University, Nanning 530004, China; 3Department of Soil, Water and Ecosystem Sciences, Indian River Research and Education Center, University of Florida—IFAS, Fort Pierce, FL 34945, USA; zhe@ufl.edu; 4Qujiang District Agricultural Technology Extension Center, Quzhou 324022, China; m13819005727@163.com

**Keywords:** cadmium contamination, soil remediation, *Vicia faba* L., agronomic traits, food security

## Abstract

Soil amendments combined with low cadmium (Cd)-accumulating crops are commonly used for remediating Cd contamination and ensuring food safety. However, the combined effects of soil amendments and the cultivation of faba beans (*Vicia faba* L.)—known for their high nutritional quality and low Cd accumulation—in moderately Cd-contaminated soils remain underexplored. This study investigates the impact of a soil amendment (SA) on agronomic traits, seed nutrition, and Cd accumulation in 11 faba bean genotypes grown in acidic soil (1.3 mg·kg^−1^ Cd, pH 5.39). The SA treatment increased soil pH to 6.0 (an 11.31% increase) and reduced DTPA-Cd by 37.1%. Although the average yield of faba beans decreased marginally by 8.74%, it remained within the 10% national permissible limit. Notably, SA treatment reduced Cd concentration in seeds by 60% and significantly mitigated Mn and Al toxicity. Additionally, SA treatment enhanced levels of essential macronutrients (Ca, Mg, P, S) and micronutrients (Mo, Cu) while lowering Phytate (Phy)/Ca, Phy/Mg, and Phy/P ratios, thus improving mineral nutrient bioavailability. Among the genotypes, F3, F5, and F6 showed the most favorable balance of nutrient quality, and yield following SA application. This study provides valuable insights into the effectiveness of SA for nutrient fortification and Cd contamination mitigation in Cd-contaminated farmland.

## 1. Introduction

Cadmium (Cd) is one of the most common and highly toxic heavy metal pollutants in agricultural ecosystems, with a half-life of 10–30 years. Cadmium accumulates in the human liver, kidneys, lungs, and bones and poses serious health risks. The International Agency for Research on Cancer has classified Cd as a Group 1 carcinogen [1]. Cadmium primarily enters the human body through the soil–crop–human chain, making soil contamination with Cd a major threat to food safety. In China, industrial emissions, agricultural activities, and climate change collectively contribute to soil Cd pollution [2,3,4,5]. The 2014 China National Soil Pollution Survey Report revealed that 16.1% of soils have excessive heavy metal content, with Cd exceeding the standard in 7.0% of cases [6,7]. Approximately 13 million hectares of farmland in China, particularly in industrialized regions such as Hunan, Guizhou, and Nanning, are affected by Cd pollution [6,7]. Yang et al. predicted through machine learning that by 2040, 41% of China’s population will be exposed to Cd levels that exceed the safety thresholds set by European and American food safety agencies [8,9]. Therefore, strategies for effective management of Cd pollution are imperative.

Soil Cd pollution can be controlled via phytoremediation, application of soil passivation agents, agronomic measures, and bioremediation [10,11,12,13,14,15,16]. However, reducing Cd accumulation in crops is a key strategy for minimizing human Cd exposure. Research indicates that the Cd accumulation capability of a crop is influenced by its genotype and soil conditions. Therefore, selecting low-Cd-accumulating crop genotypes combined with optimized soil management practices is an effective approach to reducing Cd pollution [17]. In recent years, the application of soil amendments (SAs) has become an important method for reducing Cd uptake by crops. Soil amendments alter soil properties, stabilize Cd levels, and regulate the bioavailability of nutrients, thereby promoting crop growth and mineral enrichment [10,18].

The faba bean (*Vicia faba* L.) is a versatile and productive legume known for its protein-rich seeds. Globally, faba beans are second only to soybeans in terms of production and have the highest protein content (29%) among legumes. Faba beans are highly tolerant to cold temperatures and are considered a viable alternative to soybeans [19]. China is one of the largest producers of faba beans, with the crop being often used as food and fodder. Faba beans are tolerant to waterlogging and temporary flooding, making them highly suitable for cereal and legume rotations [20,21], and for management schemes for Cd-contaminated farmland [22]. Faba bean genotypes exhibit notable differences in Cd absorption capacities, with some exhibiting a strong Cd tolerance [23,24,25,26,27]. However, research on the synergistic effects of faba bean genotypes and soil amendments, particularly in high-Cd-contaminated soils, remains limited. Systematic studies on the responses of different faba bean genotypes to soil amendments would thus provide novel solutions for Cd pollution management and food security.

In this study, we aimed to investigate the synergistic effects of an SA application and low-Cd-accumulating faba bean genotypes on Cd uptake, translocation, and accumulation; agronomic traits; and seed nutritional quality. The findings of this study are expected to provide insights into sustainable agricultural practices that enhance food safety and crop nutrition in Cd-contaminated farmlands and identify superior faba bean genotypes for use in such environments.

## 2. Results and Discussions

### 2.1. Agronomic Traits of Faba Bean Genotypes by Soil Amendment

#### 2.1.1. Theoretical Yield of Faba Bean Genotypes

In field trials, seed yield is a crucial indicator in faba bean production, as it directly affects economic value and planting decisions [28]. In this study, we assessed the yield of 11 faba bean genotypes under both control and SA conditions. The yields decreased by 8.74% under SA treatment (Figure 1A), showing a statistically significant difference (*p* < 0.05). In the CK, the average yield was 3818.99 t·ha^−1^, while under SA treatment, the average yield decreased to 3368.81 t·ha^−1^. This finding differs from the results reported by Huang et al. and Teasley et al., who found that organic amendments and biochar can increase crop yields [29,30]. According to the Technical Specifications for Immobilizing Heavy Metals in Contaminated Soil [31], the yield reductions for genotypes F2, F4, and F11 exceeded the standard by 1.69, 1.17, and 1.24%, respectively, whereas reductions for other genotypes remained below the 10% threshold, within acceptable limits [31]. Although there were variations in yields among genotypes under the two treatments, these differences were not statistically significant. Further analysis revealed that the average yields of genotypes F1, F9, F3, and F6 were significantly higher than those of other genotypes under both treatment conditions, exceeding the average yields of all tested faba bean genotypes by 36.04, 29.68, 20.38, and 7.09, respectively, demonstrating their strong yield potential.

Our findings provide practical guidance for faba bean production. Genotypes F1, F9, F3, and F6, which exhibited excellent yields, maintained high production rates even under SA treatment conditions, making them recommended varieties for farmers. These genotypes sustained high yields and demonstrated adaptability under SA treatment, making them valuable choices for optimizing faba bean planting yields and improving soil health management.

#### 2.1.2. Harvest Index (HI) and Other Agronomic Traits Adjustment

No significant differences were observed in the 100-seed weight and HI of faba beans under SA treatment compared to those of CK (Figure 1B,C). Similarly, Appendix A indicates that the plant height, number of seeds per pod, and SPAD values of mature leaves did not differ significantly between the two treatments, suggesting that these growth parameters were unaffected. Further analysis revealed that the 100-seed weight of genotypes F1, F3, and F11 was significantly higher, ranging from 59.50 to 71.51 g. Although the 100-seed weight of most genotypes was slightly reduced under SA treatment (all within 5%), the 100-seed weight of F6 and F2 increased by 2.78% (43.61 g) and 1.29% (37.19 g), respectively. This result suggests that F6 and F2 responded more favorably to SA treatment with respect to 100-seed weight, whereas F1, F3, and F11 maintained superior seed weights and economic value.

In contrast, although the HI of faba beans was insensitive to the SA treatment, dry biomass responded significantly (*p* < 0.001). Whole plant dry weights were reduced by 7.56% under SA treatment, with an average reduction of 7.43% in the shoot biomass. Appendix A shows that the disease rate decreased by 45.33% after SA treatment. This result was verified by Renaud et al. and Tao et al., who reported that different compositions of soil amendments may affect soil anurans and soil microorganisms, thus affecting the health status of crops [32,33]. Based on our results, we reasonably assumed that the reduction in biomass by specific amendments may represent an adaptive response to the environment to reduce the incidence of pests and diseases; however, additional experiments are needed to confirm this speculation. Additionally, as shown in Appendix A, faba bean stem dry biomass accounted for 34.04 to 52.04% of the total plant biomass across the 11 genotypes, suggesting that stems may represent an important site for mineral element accumulation.

### 2.2. Cd Cycling in Faba Bean Soil System in Response to Soil Amendment

#### 2.2.1. Decreased Bioavailability of Cd in Soil

After SA treatment, the pH value increased from 5.39 ± 0.07 to 6.00 ± 0.06, reflecting an 11% increase (Appendix A). Although the change between SA and CK was not statistically significant, it effectively mitigated soil acidification. By improving the soil pH, an SA treatment can mitigate the adverse effects of soil acidity on plant health and metal uptake rates. Yang et al. emphasized that strongly acidified and heavily contaminated rice soils are harmful to crop cultivation [17]. The present study demonstrated that an SA treatment had a substantial impact on improving soil properties and reducing Cd bioavailability.

The organic matter content increased from 20.98% to 23% following the SA treatment, with no statistically significant difference compared with CK. Studies by Buss et al. have shown that increased organic matter improves soil structure, water retention, and nutrient availability, and is closely correlated with soil acidity, microbial activity, and mineral absorption capacity [34,35,36,37]. A more pronounced effect was observed with regard to calcium (Ca) content; the change in total Ca content was statistically significant, showing a 59% increase while the exchangeable Ca content also increased significantly, with a 92% increase. Ji et al. and Wang et al. suggested that increased Ca content and bioavailability may alter the pH-DOC balance in the soil solution, affecting Cd uptake by crops [38,39]. In our study, increases in pH, organic matter, and Ca concentration, along with the enhanced exchangeable Ca, may have influenced the bioavailability of Cd in the soil.

Studies have shown that the concentration of DTPA-extractable Cd in soil is a key factor in determining Cd levels in plants. Exploring the effects of soil amendments on DTPA-Cd is crucial for enhancing food security [40,41]. In this study, the DTPA-extractable Cd (DTPA-Cd) concentrations decreased significantly by 37% following SA treatment (Appendix A). Lower extractable Cd levels in soil are crucial for minimizing the risk of Cd uptake by plants, which can otherwise lead to contamination of the food chain and potential health risks for consumers. These findings suggest that an SA treatment can be a valuable strategy for improving soil health and nutrient availability. The significant improvements in pH and calcium levels, and the reduction in Cd bioavailability, highlight the potential benefits of SA for crop productivity and safety.

#### 2.2.2. Reduced Cd Concentration in Faba Beans

Reducing the Cd content in the seeds was one of the key focuses of our study. As shown in Figure 2A, SA treatment reduced the average Cd content in seeds to 0.06 mg·kg^−1^, a 60% decrease from 0.15 mg·kg^−1^ under CK (*p* < 0.001), allowing the seeds to meet food safety standards. Among the 11 faba bean genotypes, F1, F2, F3, F4, and F7 had Cd levels below 0.04 mg·kg^−1^, whereas F10 contained 0.1 mg·kg^−1^, and the remaining genotypes ranged between 0.06 and 0.08 mg·kg^−1^. The reduction in Cd content varied from 33.33 to 81.25%, with F3 showing the highest reduction rate.

Similarly, SA treatment had a profound impact on Cd levels in shoot tissues compared with CK. SA treatment significantly lowered Cd levels in the pods, stems, and leaves (Figure 2A) (*p* < 0.001). Compared to CK, the Cd content decreased to 62.71, 74.36, and 81.25% under SA treatment, respectively. Given that stems constitute the largest plant biomass, their Cd levels were crucial for assessing the overall accumulation. The reduction percentages for stem Cd were highest in F7 (94.79%) and lowest in F4 (25.00%), indicating that most genotypes responded sensitively to SA treatment in terms of stem Cd accumulation, making them suitable for selection in SA-treated Cd-contaminated soils.

This study demonstrates that SA treatments significantly reduce the Cd content in the seeds, pods, stems, and leaves of faba beans, thereby enhancing crop safety in Cd-Cd-contaminated soils. The faba bean genotypes F3, F1, F2, F4, and F7 exhibited low Cd content in seeds and showed significant Cd reductions in response to SA treatment, making them excellent candidates for safe crop selection in moderately to highly Cd-contaminated fields.

#### 2.2.3. TFs and BCFs of Cd in Faba Beans

An assessment of TFs and BCFs can provide a comprehensive understanding of Cd dynamics in faba beans under SA treatment. As shown in Figure 2B, TF (seed/stem) increased from 0.22 to 0.35 (*p* < 0.01), with F7, F10, and F9 showing significant increases (0.74, 0.78, and 0.41, respectively) compared to CK. In contrast, the TF (seed/stem) of F4 decreased to 0.16, indicating an altered Cd translocation efficiency in certain genotypes. The TF (seed/leaf) also increased by 178.6%, from 0.14 to 0.39, with significant improvements in genotypes F6, F8, F9, F11, and F10, suggesting enhanced Cd translocation from leaves to seeds. However, the TF (seed/pod) increased marginally from 0.26 to 0.34 without statistical significance for SA, and TF (leaf/stem) showed variable responses among genotypes, with notable increases in F2 and F7 and decreases in F4, F6, F8, and F9 in SA. The observed increases in TFs, particularly TF (seed/stem) and TF (seed/leaf), highlight the differential Cd transport capabilities among the various genotypes. The BCF of the shoots reduced the Cd content, indicating that SA treatment effectively decreased Cd absorption and accumulation in faba beans (*p* < 0.001) (Figure 2C).

Based on the TF (seed/stem), genotypes F7, F10, and F9 were identified as optimal for Cd remediation because of their high translocation efficiency from the stem to the seed. This result may be attributed to genetic characteristics or physiological responses specific to different genotypes. The capacity for Cd mobilization from the leaves to the seeds was significantly enhanced in genotypes F6, F8, F9, F11, and F10, indicating that these genotypes were better suited for Cd accumulation. The combination of high TFs and reduced BCFs indicated that SA treatment improved the Cd translocation efficiency and underscored the genetic or physiological adaptability of different genotypes in Cd-contaminated farmland.

#### 2.2.4. Linear Correlation of Cd Concentrations in Faba Bean Tissues

To further investigate the distribution and translocation patterns of Cd in different faba bean tissues, we conducted a correlation analysis of Cd concentrations in the seeds, pods, stems, and leaves of 11 faba bean genotypes under CK and SA conditions (Figure 3). The results revealed highly significant linear correlations (*p* < 0.001) between the Cd concentrations in the edible parts of faba beans and those in the pods, stems, and leaves. Specifically, the correlation coefficients *R*^2^ between the Cd concentration in the seed and the Cd concentrations in the seed, pod stem, and leaf were 0.70371, 0.59545, and 0.48515, respectively. The highest regression coefficient (0.19379), which was observed for the pod, indicates the strongest linear correlation of the Cd concentration between seed and pod. This result suggests that Cd accumulation in the pods most significantly affected the seed Cd concentration, followed by the leaves and stems. Consequently, Cd accumulation in faba bean seeds was influenced by multiple tissues rather than an individual tissue. Thus, comprehensively controlling Cd accumulation across different tissues may more effectively reduce seed Cd concentrations and improve crop safety.

#### 2.2.5. Shoot Cd Uptake in Faba Beans

Figure 1D clearly illustrates the dry biomass of various aboveground tissues of the faba bean genotypes. By integrating the results from Figure 2A, we accurately calculated the Cd uptake for each genotype under SA treatment and control conditions using the total accumulation of the Cd analysis formula (Figure 4A) and precisely determined the proportion of Cd uptake in each tissue (Figure 4B). Figure 4A demonstrates that all tested faba bean genotypes exhibited significant differences in total Cd uptake in shoots under SA treatment compared to the control (*p* < 0.001 for all genotypes). This result suggests that SA treatment significantly affected Cd uptake and distribution in faba beans and potentially played a crucial role in regulating Cd transport mechanisms. Specifically, under control conditions, the total shoot Cd uptake across all genotypes ranged from 2.54 to 4.96 g·ha^−1^ (with an average of 4.09 g·ha^−1^). However, under SA treatment, the total Cd uptake was reduced by 75%, with a range of 0.63 to 1.51 g·ha^−1^ (average of 1.10 g·ha^−1^). The genotypes F7, F3, and F6 showed the most pronounced reductions, with decreases of 90, 80, and 79%, respectively.

Figure 4B shows the Cd uptake distribution across different shoot tissues in the 11 faba bean genotypes under CK and SA treatment. In CK, stems were the main site of Cd uptake, accounting for 34% to 68% of the uptake (with an average of 55%). Under SA treatment, stems remained the primary site, with an average uptake of 50%, representing an 5% decrease compared to their uptake when receiving CK treatment. However, there were notable differences in the different genotypes’ responses to SA treatment. Specifically, genotypes F7, F10, F3, and F1 exhibited a significant reduction in the percentage of Cd in the stem, whereas F4 and F8 showed a significant increase, and the other genotypes showed no significant change. This result indicates that although SA treatment effectively reduced the overall Cd uptake and accumulation by modulating Cd transportation, the distribution patterns of Cd within faba beans remained generally stable, with significant variations in Cd distribution mechanisms among different genotypes.

SA treatment significantly reduced Cd uptake in the shoots of faba beans, with genotypes F7, F3, and F6 showing the most pronounced responses, and affected the tissue-specific distribution of Cd across the different genotypes (Figure 4). For all tested faba bean genotypes, regardless of whether CK or SA treatment was administered, the stem was generally the primary site of Cd accumulation. However, in certain genotypes, the proportion of Cd in the stem varied significantly under SA treatment. This result suggests that although SA treatment reduced shoot Cd accumulations by modulating Cd transport mechanisms, its impact on Cd distribution differed among genotypes. These findings suggest that SA treatment can potentially improve the management of Cd content in faba beans by optimizing Cd transport and distribution mechanisms, as well as provide insights into optimizing Cd management strategies tailored to specific genotypes. This finding has important implications for the breeding of faba bean genotypes with enhanced Cd tolerance and regulatory capacity.

### 2.3. Absorption and Distribution of Mineral Elements in Faba Bean Seeds by Soil Amendment

#### 2.3.1. Correlation Between Elements and Yield in Seeds

In this study, SA treatment clearly and significantly reduced the Cd concentration in faba bean seeds; however, the yield was also slightly reduced. Therefore, we hypothesize that SA treatment may affect the uptake and accumulation of other elements in faba bean seeds, thereby affecting yields. As shown in Figure 5A and Appendix A, CK and SA were divided into two clusters, which reflects the unique role of SA treatment in regulating element accumulation and crop performance. In addition, SA treatment positively affected the accumulations of Ca, S, Mo, P, Mg, and Cu, whereas it had no effect or even an inhibitory effect on the accumulations of Fe, Zn, Co, Al, Cd, and Mn. The six elements Ca, S, Mo, P, Mg, and Cu showed similar behavioral patterns in response to SA treatments, which may indicate their synergistic effects or interdependence in uptake, transportation, or metabolism in plants. Trace elements such as Fe, Zn, Co, Al, Cd, and Mn were clustered with yields, implying that the accumulation levels of these elements may have jointly influenced the key physiological processes of faba bean growth and yields and that changes in these elements might cause major changes in yields under an SA treatment.

#### 2.3.2. Modulation of Elemental Concentrations in Seeds

Figure 5B shows significant increases in Ca, Mg, P, S, Mo, and Cu in seeds under SA treatment, whereas Al, Cd, and Mn decreased significantly. The Fe, Zn, and Co levels remained unchanged. Specifically, the macroelements Ca, Mg, P, and S increased by 21.26%, 13.50%, 12.94%, and 15.77%, respectively, while Mo and Cu increased by 544.42% and 13.62%. Cd, Mn, and Al exhibited reductions of 59.63%, 48.70%, and 24.68%, respectively. Mn decreased by 47.54% in line with Cd reduction. Mo content increased between 271.26% and 918.09%, indicating enhanced Mo uptake and translocation. Cu content varied among genotypes, with decreases in F5, F9, and F10, whereas Fe, Zn, and Co responses were genotype-specific. Notably, Zn increased in F11, F2, and F3, whereas Co increased in F3, F1, and F11. Fe contents increased significantly in F6, F5, F3, and F2.

Overall, SA treatment effectively improved the nutrient content of faba bean seeds through genotype-specific responses. F11 combined with SA is suitable for environments that require high Zn and Co concentrations, whereas F6 combined with SA is ideal for iron-deficient areas. Monitoring Mo levels is crucial to prevent interference with the absorption of other nutrients.

### 2.4. Nutritional Quality and Mineral Bioavailability in Seeds

SA treatment exhibited significant effects on mineral regulation but also led to a slight reduction in seed yield. The decrease in yield suggests that SA treatment may influence other key seed components such as proteins and phytic acid. Proteins, as a core nutritional component of faba beans, directly affect their nutritional value, whereas phytic acid, as a major form of phosphorus storage and an anti-nutritional factor, affects mineral absorption and utilization. Given that SA treatment significantly increased the levels of P and S in seeds, it was crucial to assess changes in phytic acid and protein content to comprehensively evaluate the impact of SA treatment.

Despite the significant increase in P and S levels, the protein and phytic acid content did not exhibit significant changes across all genotypes under SA treatment (Figure 6A). This indicates that SA treatment primarily affected the accumulation of specific minerals rather than the synthesis or accumulation of proteins and phytic acid, possibly because of the inherent stability of the biosynthetic pathways for these components. This stability helps maintain the nutritional quality of the seeds, even with variations in mineral content. However, for specific genotypes, such as F5 and F11, notable changes were observed: phytic acid content in the F5 genotype decreased by 64% and protein content in the F11 genotype decreased by 21% under SA treatment, which may be attributed to the genotype-specific responses of faba beans.

The ratios of phytic acid to minerals were analyzed to further understand the effect of SA treatment on nutritional quality. Under SA treatment, the Phy/Ca (Figure 6B,C), Phy/Mg (Appendix A), and Phy/P (Figure 6D,E) ratios in faba bean seeds were significantly reduced by approximately 23, 20, and 19%, respectively (all *p* < 0.05). However, compared with the CK treatment, the Phy/Fe (Appendix A) and Phy/Zn (Appendix A) ratios showed no significant differences under SA treatment, with a 3% increase and an 8% decrease, respectively. As shown in Figure 6C,D, SA treatment significantly reduced the Phy/Ca ratio in the F5 genotype by 76%, and the Phy/P ratios in the F6, F5, and F8 genotypes by 71%, 61%, and 48%, respectively. Additionally, Appendix A shows reductions of 65% and 60% in the Phy/Mg ratios for the F6 and F5 genotypes under SA treatment. These findings suggest that SA treatment significantly reduced the phytic acid–mineral ratios (e.g., calcium, magnesium, and phosphorus) in specific faba bean genotypes like F5 and F6, potentially enhancing mineral bioavailability. This indicates that targeted SA application could improve mineral absorption efficiency without notably affecting protein or phytic acid content, enhancing the nutritional value of faba beans. These insights open up new possibilities for developing nutritionally improved faba bean varieties and present promising directions for crop improvement and agricultural production.

## 3. Materials and Methods

### 3.1. Materials and Field Experiments

We conducted a field experiment in moderately Cd-contaminated acidic soil to evaluate the performance of 11 faba bean genotypes treated with SA. The field experiment was conducted in Cd-contaminated agricultural land in Lijin Garden Village, Quzhou City, Zhejiang Province. The study employed a randomized complete block design with three replicates to ensure robust comparisons. The experiment included both a control group and a treatment group. The treatment group was supplied with a soil amendment (SA), Greenlife No. 1 [42] (Purchased from Greenlife Technologies, Inc., Hangzhou, China), at a rate of 6000 kg·ha^−1^ alongside basic fertilizers, while the control group received only basic fertilizers without any soil amendment. This approach allowed us to comprehensively assess the effects of soil amendment on crop performance under moderately Cd-contaminated conditions. All tested faba bean genotypes were sourced from local seed companies in Zhejiang Province, China, and are detailed in Appendix A and Appendix A. The genotypes, referred to as F1, F2, …, F11, were obtained from markets in Hangzhou and are commonly used by Chinese farmers. Specifically, we examined changes in soil properties, Cd bioavailability, crop yields, and concentrations of essential macronutrients and micronutrients in faba bean seeds. This study aimed to evaluate the effects of soil amendment supply on faba bean cultivation in moderately Cd-polluted soil. Further details regarding the materials and field experiments are provided in Appendix A.

### 3.2. Plant Samples

#### 3.2.1. Yield and Biomass Analysis

All genotypes of faba beans were grown for 220 days in moderately Cd-contaminated farmland, and after harvest, estimates of theoretical seed yield and disease rates were calculated. Following classification and analysis, the dry weights of the plants were measured, and relevant agronomic indices and mineral content of different tissues, as well as the nutritional composition of seeds, were determined. The specific experimental procedures for analyzing the above indicators in the laboratory are detailed in Appendix A.

#### 3.2.2. Metal Element Analysis

The acid digestion method was used to digest all plant tissues. Macroelements were analyzed by ICP-OES and microelements by ICP-MS. The digestion and determination processes for the metal elements are detailed in Appendix A.

#### 3.2.3. Analyses of Protein and Phytic Acid Contents and Phy/M

The crude protein content in the faba bean seeds was determined via the classical Kjeldahl nitrogen determination method to obtain the nitrogen content of the seeds [43]. The crude protein content was calculated by multiplying the nitrogen content by a conversion factor of 6.25 [44]. This method ensures an efficient and accurate determination of crude protein content in seed samples.

The phytic acid content of faba bean seeds was determined using the ferric chloride colorimetric method and the absorbance value of the solution to be tested was measured at 500 nm using an enzyme marker [43,45]. The specific operation and measurement process can be found in Appendix A.

The formulas for phytate–mineral molar ratios (Phy/M) in the seed of faba bean plants are as follows:$$Phy/M=\left(C_{Phy}/660\right)/\left(C_{M}/M_{r}\right)$$

*C_Phy_* (mg·kg^−1^) and *C_M_* (mg·kg^−1^) denote the contents of phytic acid and mineral elements in faba bean seeds; *M_r_* indicates the mineral element.

#### 3.2.4. Analysis of Bioaccumulation Factor, Translocation Factor, and Total Cadmium Accumulation

The formulas used to calculate the bioaccumulation factor (BCF), translocation factor (TF), and total accumulation of Cd (TA_Cd_, g·ha^−1^) in the aboveground parts of the faba bean plants were as follows:$$BCF=C_{up}/C_{soil}$$
$$TF=C_{up}/C_{low}$$
$${TA}_{Cd}=\sum {Cs}_{Cd}\times {Biomass}_{n}$$

*C_up_*, *C_low_*, and *C_soil_* denote the Cd concentrations in the upper and lower tissues of faba bean, as well as in the soil, respectively. *Cs_Cd_* indicates the concentration of Cd in the shoots of faba beans and *Biomass_n_* refers to the biomass (g·ha^−1^) across different aboveground faba bean tissues.

### 3.3. Soil Sample Collection and Physicochemical Property Analysis

Soil samples were collected at a depth of 10–20 cm from three areas in the CK and SA groups using a stainless-steel shovel. After sampling using the five-point plum throat method, the samples were mixed, air-dried, ground, and sieved through a 100-mesh screen, resulting in three pre-planting soil samples and six post-harvest faba bean soil samples (three from CK and three from SA) that were stored in sealed plastic bags [43]. The moderately Cd-contaminated soil was a brown loam, and the basic properties of the soil were determined following the risk control standards for environmental quality of contaminated soil in agricultural soil in China, as shown in Appendix A. The specific determination and analysis of soil collection and the soil’s physicochemical properties are described in Appendix A.

## 4. Statistical Analysis

All data are presented as the mean ± standard deviation (SD) of the three replicates. Statistical analysis was performed using the IBM SPSS Statistics software (version 26.0; one-way ANOVA followed by Tukey’s post hoc test (with a significance level of *p* < 0.05) was used to determine significant differences between treatment groups. Data were visualized using the Origin 2023 software. In all figures and tables, unless otherwise specified, data are presented as a mean ± SD, with three replicates per treatment, and different uppercase and lowercase letters indicate significant differences between faba bean genotypes (*p* < 0.05). The significance levels between SA and CK treatments are denoted as follows: *** for *p* < 0.001, ** for *p* < 0.01, * for *p* < 0.05, while “ns” and non-marked differences in figures indicate no significant difference.

## 5. Conclusions

In this study, we used the strategy of soil amendment (SA) intercropping with a Cd-low-accumulating crop, faba bean (*Vicia faba* L.), to address food security issues in Cd-contaminated farmland. The application of SA was effective in reducing the bioavailability of Cd, increasing the acidity of the soil, and significantly reducing Cd accumulation in faba bean seeds, ensuring that most of the tested genotypes of faba beans met national food security standards. The application of SA significantly increased the essential mineral nutrients (Ca, Mg, P, and S) and their bioavailability, increased the micronutrients Mo and Cu, and reduced Mn and Al accumulations. There was a slight decrease in faba bean yields, which remained within the acceptable range of national standards. In this study, three faba bean genotypes, F3, F5, and F6, had the most balanced combined performance in terms of nutrient quality, food security, and yield in moderately Cd-contaminated soils in synergy with SA. These findings provide valuable insights into mitigating Cd contamination in faba beans using SA, thereby contributing to safer food production and improved nutrient composition in Cd-contaminated farmland. However, further research is needed to confirm these findings across various soils and environments, assess long-term effects on crop yields and quality, and explore interactions with other soil amendments. Overall, this study offers a valuable foundation for developing sustainable strategies to manage cadmium contamination and enhance nutrient-rich crop production.

## Figures and Tables

**Figure 1 plants-14-00141-f001:**
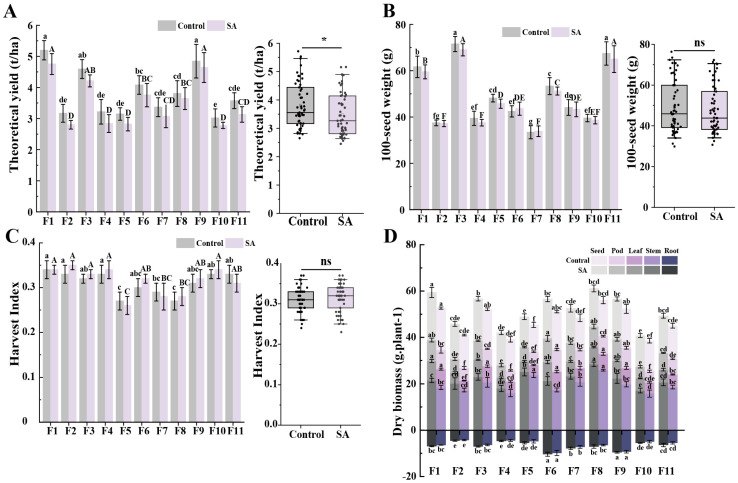
(**A**) Theoretical yield, (**B**) 100-seed weight, (**C**) harvest index (HI), and (**D**) dry biomass of 11 faba bean genotypes under control and SA treatments. Significant differences at *p* < 0.05 between groups are indicated by different letters. * represents a significant difference at *p* < 0.05 compared to the control, and “ns” denotes no significant difference.

**Figure 2 plants-14-00141-f002:**
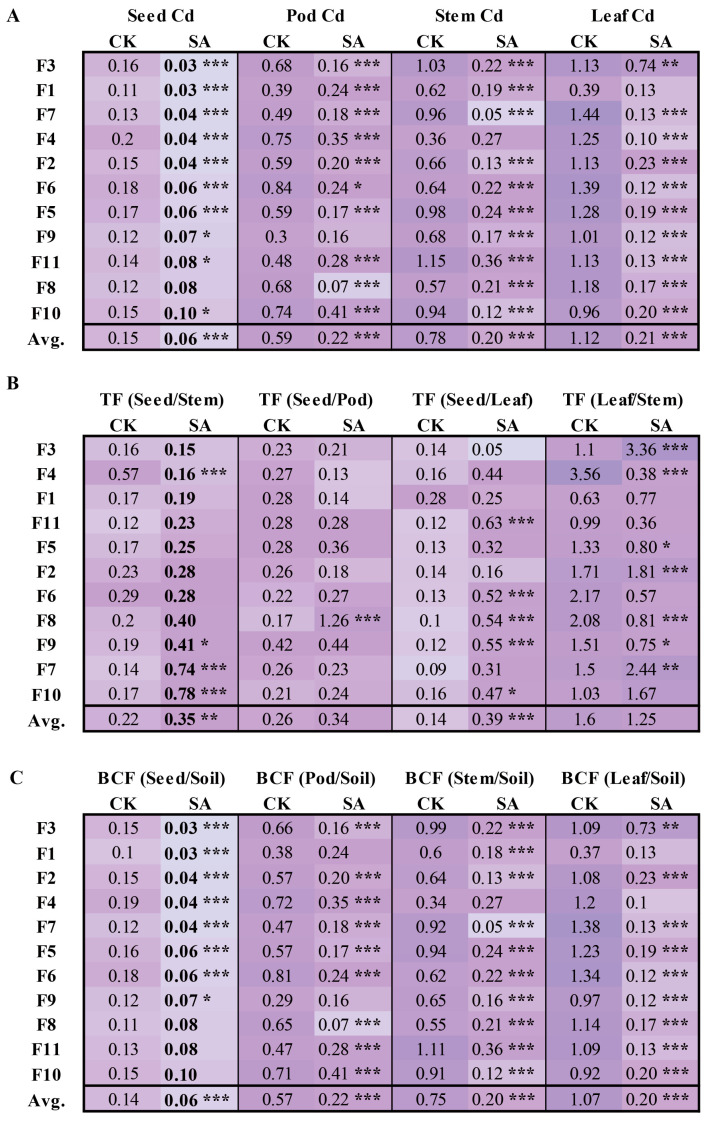
(**A**) Cadmium (Cd) concentrations in the aerial tissues of faba beans, including seed, pod, stem and leaf; (**B**) translocation factor (TF) and (**C**) BCFs of Cd in the aerial parts of faba beans. The significance between SA and CK treatments is as follows: *** for *p* < 0.001, ** for *p* < 0.01, * for *p* < 0.05, and no marking indicates no significant difference. The light purple color indicates the corresponding smaller value and the dark purple color indicates the corresponding larger value.

**Figure 3 plants-14-00141-f003:**
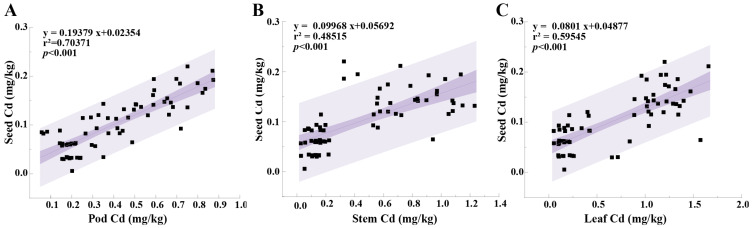
Linear correlation analysis of Cd concentrations in faba bean tissues. The solid line represents the best-fit linear regression and the shaded area indicates the 95% confidence interval. Sample sizes for (**A**–**C**) are N = 66 each.

**Figure 4 plants-14-00141-f004:**
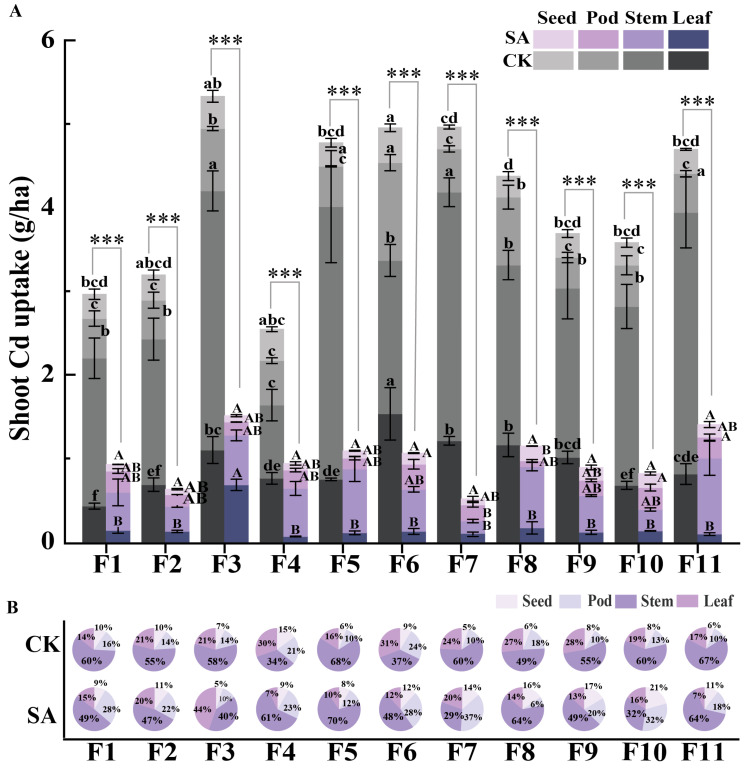
Comparison of Cd uptake and distribution in the shoots of 11 faba bean genotypes under SA and CK treatments. (**A**) Shoot Cd uptake (g·ha^−1^) and its significant differences. (**B**) Percentage of cadmium uptake in four tissues (seed, pod, stem, leaf) within the shoots. Groups marked with different letters exhibit a significant difference at *p* < 0.05. *** indicate a significant difference (*p* < 0.001) when compared to CK.

**Figure 5 plants-14-00141-f005:**
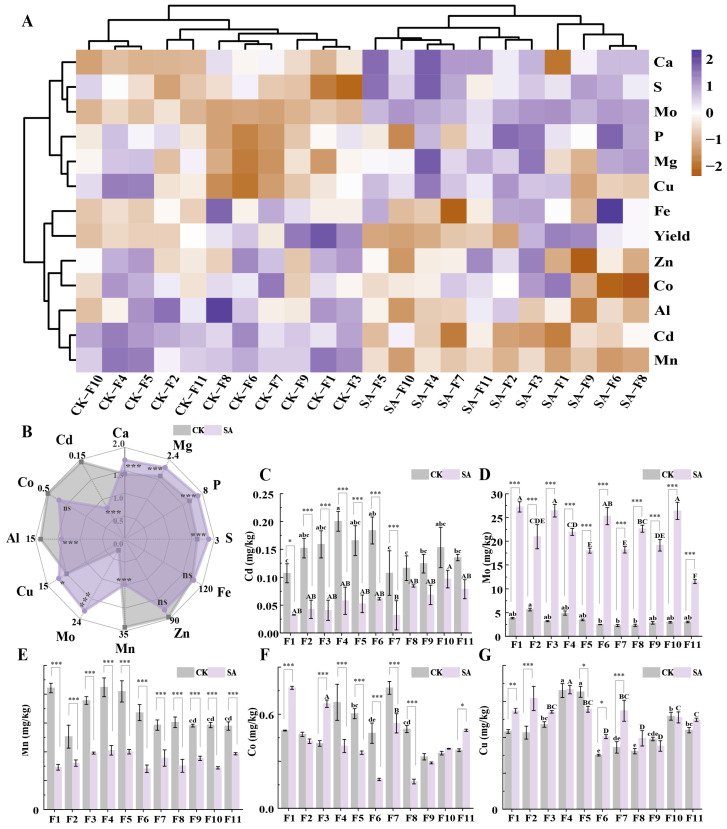
Correlation analysis between elemental contents and yield, and modulations in elemental concentrations in faba bean seeds after SA treatment. (**A**) Clustered heatmap of seed metal elements and yield in CK and SA treatment for all tested faba bean genotypes. (**B**) Radar chart comparing all elements’ concentrations after SA treatment vs. CK, with significance marked within the purple area. (**C**–**G**) Concentrations of the trace elements Cd, Mo, Mn, Al, and Cu in seeds, each represented individually. Gray represents CK, and pink and purple represent SA. Ca, Mg, P, and S are in g·kg^−1^; Fe, Zn, Mn, Mo, Cu, Al, Ni, Co, and Cd are in mg·kg^−1^. Significant differences at *p* < 0.05 between groups are indicated by different letters. *, **, *** represent, respectively, significant differences at *p* < 0.05, *p* < 0.01, and *p* < 0.001 compared to CK, while non-marked differences represent no significance.

**Figure 6 plants-14-00141-f006:**
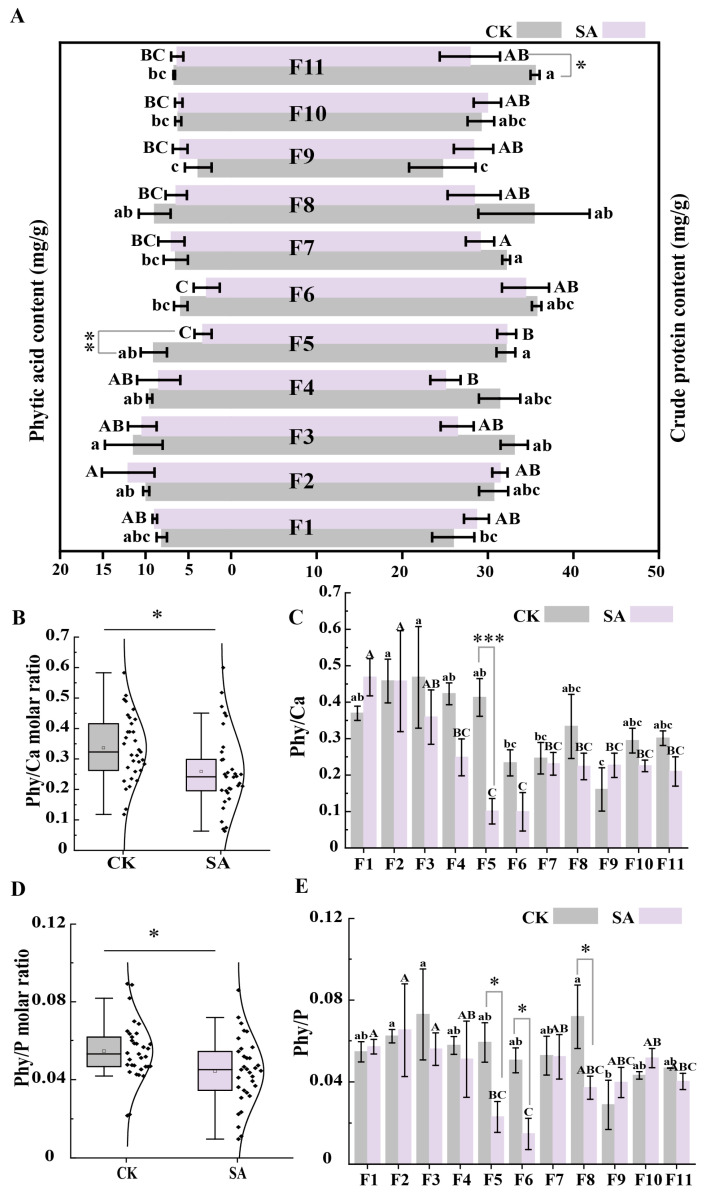
Protein content, phytic acid content, and mineral bioavailability (phytic acid/mineral ratio) in faba bean genotypes. (**A**) Crude protein content (mg·g^−1^) and phytic acid content (mg·g^−1^) in 11 faba bean genotypes under SA treatment compared to CK. (**B**,**C**) depict the response of the Phy/Ca molar ratio in all faba bean genotypes to SA treatment. (**D**,**E**) illustrates the response of the Phy/P molar ratio under the same SA treatment. Different letters signify significant differences between groups at *p* < 0.05. * indicates a significant difference at *p* < 0.05, ** at *p* < 0.01, *** at *p* < 0.001, and non-marked differences represent no significance.

## Data Availability

This manuscript does not report data generation or analysis. Data are available from the authors upon request.

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
