# Peer review of "Evaluating a Soil Amendment for Cadmium Mitigation and Enhanced Nutritional Quality in Faba Bean Genotypes: Implications for Food Safety"

_plants, 2025, doi:10.3390/plants14010141_

Round 1

Reviewer 1 Report

Comments and Suggestions for Authors

Dear Authors, you should address my comments highlighted across the text and the supplementary file.

Author Response

Response to Reviewer 1's Comments

We sincerely appreciate the time and effort the reviewers have dedicated to providing thoughtful and constructive feedback on our manuscript. Your insightful suggestions have been invaluable in enhancing the scientific rigor, clarity, and overall quality of our work. Below, we provide a detailed point-by-point response to each of your comments, outlining the changes made to the manuscript. We would like to inform you that all the changes made to the manuscript in response to the reviewer comments have been marked in red for easy identification. This will allow you to quickly review the revisions and ensure that all suggestions have been addressed appropriately. We sincerely thank you for your detailed evaluation and valuable contributions, which have greatly improved the quality, clarity, and overall quality of the manuscript.

Comments 1: Line 28, Please, amend as shown above across the text. “mg.kg-1”  

Response 1: Agree. Thank you for your meticulous review and suggestions. We have revised all instances of "mg/kg" to "mg·kg⁻¹" to align with international standards. These changes have been carefully implemented and are marked in red in the revised manuscript. Specifically, the modifications are located on Line 28, Lines 171–174, Line 302, and Line 403, totaling nine revisions. We sincerely appreciate your attention to detail, which has significantly contributed to enhancing the professionalism and academic rigor of the article.

Comments 2: Line 33, Please, amend as shown above across the text. “phytate (Phy)”

Response 2: Agree. Thank you for your valuable suggestion. We have revised the text to ensure clarity and consistency. Specifically, the first occurrence of the abbreviation has been updated to "phytate (Phy)/Ca" to align with the standard notation, and the changes are marked in red in the revised manuscript. The subsequent ratios, such as "Phy/Mg" and "Phy/P," will remain in their original format. We appreciate your attention to detail, which has greatly contributed to improving the precision and readability of our work.

Comments 3: Line 38, Please, replace these words with others different from those included in the title.

Response 3: Agree. Thank you very much for your insightful suggestion. After careful consideration and internal discussion within our research team, as well as reviewing relevant studies, we have decided to replace the terms "soil amendment" and "faba beans" with more appropriate alternatives. Specifically, "soil amendment" has been replaced with "soil remediation", and "faba beans" has been replaced with "Vicia faba L.", and the changes are marked in red in the revised manuscript. We sincerely appreciate your attention to detail, which has contributed significantly to improving the clarity and precision of our manuscript. Your valuable input has greatly helped us enhance the quality of our work. Thank you once again for your thoughtful suggestions.

Comments 4: Line 91, Please, describe the results mentioning the statistical significance across this section.

Response 4: Agree. Thank you for your thoughtful comment. We have carefully considered your suggestion and revised the description of the results to explicitly mention the statistical significance. Originally, our consideration was that, since significance is usually indicated by the "*" symbol in the figures, we typically do not emphasize it again in the text. However, recognizing that the yields represent a critical piece of data in this study, we agree that it is important to specifically highlight the statistical significance of the 8.74% decrease in yields under the SA treatment. We have updated the representation of statistical significance to ensure it aligns with more formal academic standards. The revised sentence now reads: “The yields decreased by 8.74% under SA treatment (Figure 1A), showing a statistically significant difference (p < 0.05).” Thank you again for your insightful suggestion, which has helped us improve the clarity and accuracy of our presentation.

Comments 5: Line 92, Please, amend as shown above across the text.” t. ha-1”

Response 5: Agree. Thank you for your insightful comment. Based on your suggestion, we carefully reviewed the manuscript and made comprehensive revisions to ensure the correct and consistent use of notation formats. Specifically, we have implemented the following updates:

Line 91 and Line 92: "t/ha" has been revised to "t·ha⁻¹".

Line 248, Line 249, Line 278, Line 409, and Line 416: "g/ha" has been revised to "g·ha⁻¹".

Line 375: "kg/ha" has been revised to "kg·ha⁻¹".

Line 302: "g/kg" has been revised to "g·kg⁻¹".

Line 359: "mg/g" has been revised to "mg·g⁻¹".

We sincerely appreciate your attention to these details. It has allowed us to identify and correct inconsistencies, ensuring the manuscript adheres to higher academic and professional standards. These revisions have been carefully incorporated and are marked in red in the updated manuscript. Thank you once again for your valuable guidance.

Comments 6: Line 113, Please, add the meaning of the letters. Do the same in the other Figures across the text.

Response 6: Agree. Thank you very much for your thoughtful and constructive suggestion. We sincerely appreciate your attention to detail. In response, we have now included the explanation of the statistical significance symbols in the figure legends for Figure 1, Figure 2, Figure 4, Figure 5, and Figure 6, with the modifications marked in red in the revised manuscript.

We would like to apologize for any confusion caused. Our original intention was not to overlook this aspect, but rather to avoid redundancy in the manuscript. We believed that the explanation of statistical symbols in the "4. Statistical analysis" section (Lines 429–439) was sufficient to cover all figures.

Your suggestion has been extremely valuable and has contributed greatly to making the manuscript more consistent and rigorous in terms of formatting. We truly appreciate your careful and helpful review. Thank you once again for your insightful input.

Comments 7: Line 114, The first letter of the words included in the Titles must always be capital.

Response 7: Agree. Thank you very much for pointing out this issue so carefully. Upon reflection, we feel deeply regretful and recognize that our oversight on the capitalization of the first letters in titles reflects a lack of attention to detail in our work. For this, we sincerely apologize and truly appreciate your valuable suggestion.

We have carefully reviewed all the subtitles and made corrections to ensure that the first letter of each word is capitalized. These revisions have been marked in red in the revised manuscript. Your suggestion has not only improved the overall precision and compliance of our manuscript but also reminded us of the importance of paying closer attention to such details in academic writing. Moving forward, we will take this as a valuable lesson and hold ourselves to a higher standard to prevent similar issues in the future. Once again, we greatly appreciate your patience and guidance.

Comments 8: Line 364-366, Please, move this paragraph at the beginning of the sub-section 3.1. and report the synthetic description of the experimental protocol with the chosen design, prior to provide further details related to cultivar and soil amendment.

Response 8: Agree. We greatly appreciate your insightful suggestion, and we completely agree with your opinion. Your recommendation is of significant importance to us, as it has guided us in improving the structure and clarity of this section. Following your suggestion, we have moved the paragraph to the beginning of subsection 3.1. and included a concise description of the experimental protocol and chosen design before providing further details about the faba bean genotypes and soil amendment treatments. These changes have enhanced the scientific rigor, detail, and logical flow of the 3.1. Materials and Field Experiments subsection. Thank you once again for your valuable input, which has been instrumental in refining our manuscript. All revisions have been marked in red for your reference.

Comments 9: Line 435, modifying the commas (',') to ', and'.

Response 9: Agree. Thank you for your valuable suggestion. We fully agree with your recommendation. In response, we have revised Line 435 by changing the comma (",") to ", and" as requested. We appreciate your attention to detail, and your feedback has been instrumental in improving the clarity and accuracy of the manuscript. We also acknowledge that our command of English grammar is still developing, and we are grateful for your constructive feedback to help us improve.

Comments 10-11: Line 442, “the faba bean”delete “the” and "Vicia faba" needs to be italicized.

Responses 10 and 11: Agree. We appreciate your meticulous feedback and strongly agree with these suggested changes. On Line 442, we have deleted “the” before “faba bean” as recommended. We also recognize the importance of grammatical accuracy and will continue to improve our writing skills to ensure better adherence to grammatical standards in the future. The scientific name of faba bean, Vicia faba L., has been italicized throughout the manuscript to adhere to proper formatting standards for botanical nomenclature. We sincerely thank you for pointing out the inconsistency in the formatting of the scientific name Vicia faba L. in our manuscript. Your feedback has been invaluable in improving the accuracy and professionalism of our work.

Comments 12-14: In the 5. Conclusions section: Line 449, delete the phrase "but the reduction in yield was"; Lines 454-455, amend "However, the single growing season and limited genotypes tested, restrict its generalizability. Further" to "However, further".

Response 12:

Agree. Thank you for your careful review and thoughtful suggestion. We agree with the proposed change and have deleted the phrase "but the reduction in yield was" from Line 449.

Responses 13 and 14Agree. We sincerely appreciate your insightful comments. We have revised the sentence on Lines 454-455 to read: "However, further studies are needed to validate the findings." This adjustment enhances the grammatical correctness and clarity of the conclusions. Your feedback has been invaluable in improving the manuscript, and we are grateful for your attention to detail and constructive suggestions.

We sincerely appreciate your thorough and constructive feedback. The suggestions have been instrumental in refining our manuscript. We have carefully revised the manuscript to address each point raised, and we believe these changes have improved the clarity, coherence, and scientific rigor of the work. All revisions have been marked in the manuscript for easy reference. Thank you once again for your valuable contributions, which have greatly enhanced the quality of our research. Your input has been indispensable, and we are grateful for the time and effort you have invested in reviewing our manuscript.

Reviewer 2 Report

Comments and Suggestions for Authors

This manuscript provides novel information on the faba bean grown in Cd-contaminated farmland. Soil amendment used reduced the bioavailability and accumulation of Cd in faba bean seeds, but there were genotypical response differences. This was also valid for the essential seed mineral nutrients (Ca, Mg, P, and S). While application increased also some micronutrients (Mo and Cu), it negatively affected the accumulation of Mn and Al.

Yet, the data is from a single growing season and location.  Findings should be confirmed via trials at different soil types and different environmental conditions. 

Considering the novelty of the information, this reviewer believes the findings will be interesting for the readers of the  'Plants' journal. Manuscript may trigger further research on the topic. 

The following minor concerns are denoted:

- Please remove 'food security from line 35.

- Line 66 'Vicia faba' should be in Italic.

- Lines 73-76. A sentence is repeated. Please correct also the cited references.

- This reviewer suggests citing & discussing the following directly relevent manuscripts:

Shi, L., et al. (2022). "Effects of combined soil amendments on Cd accumulation, translocation and food safety in rice: a field study in southern China." Environmental Geochemistry and Health 44(8): 2451-2463.

Wang, Y. M., et al. (2020). "Effect of amendments on soil Cd sorption and trophic transfer of Cd and mineral nutrition along the food chain." Ecotoxicology and Environmental Safety 189.

Author Response

Response to Reviewer 2's Comments

We would like to express our sincere gratitude for your thoughtful and constructive feedback on our manuscript. Your comments have been invaluable in improving the clarity and rigor of our study, marked in red. We appreciate the time and effort you took to review our work, and we are pleased that you found our findings novel and potentially impactful for further research in this area. Below, we will carefully address each of your comments point by point.

Comments 1: This manuscript provides novel information on the faba bean grown in Cd-contaminated farmland. Soil amendment used reduced the bioavailability and accumulation of Cd in faba bean seeds, but there were genotypical response differences. This was also valid for the essential seed mineral nutrients (Ca, Mg, P, and S). While application increased also some micronutrients (Mo and Cu), it negatively affected the accumulation of Mn and Al.

Yet, the data is from a single growing season and location.  Findings should be confirmed via trials at different soil types and different environmental conditions.

Considering the novelty of the information, this reviewer believes the findings will be interesting for the readers of the 'Plants' journal. Manuscript may trigger further research on the topic.

Response 1: We sincerely appreciate you for recognizing the novelty of our work and for their positive evaluation of the study’s contribution. We are grateful for your appreciation of the findings presented in this manuscript.

Regarding the primary concern you raised: “However, these data come from a single growing season and location. Further trials in different soil types and environmental conditions are needed to confirm the results,” we fully understand and acknowledge the importance of this issue. We recognize your concern about the generalizability and robustness of the study’s findings. In our study design, we selected typical Cd-contaminated agricultural soils to reflect real-world conditions, which we believe makes the study relevant and applicable to practical agricultural settings. Our primary focus was on the response of different genotypic varieties combined with soil amendments in typical Cd-contaminated farmland. Similar studies, such as those by Wan X. et al.(2024, STE) and Lian J. P. et al. (2023, Food Chemistry), also employed a single-site design, and they conducted comparable research in a specific location. This approach helps to draw reliable initial conclusions within the context of the experimental conditions and provides a valuable theoretical foundation for subsequent research. That being said, we do acknowledge that basing the study on a single growing season and location may limit the broader applicability of the results. To address this limitation, we have highlighted in the “5. Conclusions” section: "However, further research is needed to confirm these findings across various soils and environments, assess long-term effects on crop yields and quality, and explore interactions with other soil amendments." This statement acknowledges the potential limitations associated with the single growing season and location and indicates the need for future studies to confirm the findings.

We wholeheartedly agree with your assessment, and we deeply value your scientific rigor in pointing out these considerations. We sincerely appreciate your constructive feedback, which not only helped us better recognize the limitations of this study but also provided valuable guidance for future research design. Looking ahead, we plan to conduct further trials in diverse soil types and environments to verify and expand upon the conclusions drawn in this study. We hope that the current research will contribute useful insights to the field and inspire more in-depth studies on Cd-contaminated soil remediation and crop nutritional improvement. Once again, thank you for your invaluable suggestions.

Comments 2:

The following minor concerns are denoted:

  1. Please remove 'food security from line 35.
  2. Line 66 'Vicia faba' should be in Italic.
  3. Lines 73-76. A sentence is repeated. Please correct also the cited references.
  4. This reviewer suggests citing & discussing the following directly relevent manuscripts:

Shi, L., et al. (2022). "Effects of combined soil amendments on Cd accumulation, translocation and food safety in rice: a field study in southern China." Environmental Geochemistry and Health 44(8): 2451-2463.

Wang, Y. M., et al. (2020). "Effect of amendments on soil Cd sorption and trophic transfer of Cd and mineral nutrition along the food chain." Ecotoxicology and Environmental Safety 189.

Response 2:We appreciate the thoughtful and constructive feedback provided by the reviewer. Below, we have addressed each comment in detail and made the necessary revisions to improve the manuscript accordingly.

  1. Comment: Please remove 'food security' from line 35.

Response: Agree. Thank you for pointing this out. We have removed the term "food security" from line 35 as requested. After carefully considering the term "food security" based on your feedback, we have concluded that "food security" is a broad concept encompassing multiple dimensions, including food availability, accessibility, nutritional adequacy, and stability. Our study primarily focuses on nutrient quality and yield, which are specific and measurable indicators. Since the broad concept of food security already encompasses nutrient quality and yield, reiterating "food security" may reduce the precision and scientific rigor of the statement. Therefore, we have removed "food security" from line 35 to ensure the manuscript remains more focused and rigorous. Thank you again for your insightful suggestion.

  1. Comment: Line 66 'Vicia faba' should be in Italic.

Response: Agree. We apologize for this oversight. We have corrected the formatting of 'Vicia faba' on line 66 to italicize it, in accordance with the standard convention for scientific names. Additionally, we have reviewed the manuscript and made the necessary corrections to all instances of 'Vicia faba' throughout the text. Thank you again for your helpful suggestion.

  1. Comment: Lines 73-76. A sentence is repeated. Please correct also the cited references.

Response: Agree. We appreciate your attention to this issue. We sincerely apologize for this oversight. We have removed the repeated sentence in lines 73-76 and have updated the corresponding references accordingly. We appreciate your understanding and helpful suggestion.

  1. Comment: This reviewer suggests citing and discussing the following directly relevant manuscripts:

Shi, L., et al. (2022). "Effects of combined soil amendments on Cd accumulation, translocation and food safety in rice: a field study in southern China." Environmental Geochemistry and Health 44(8): 2451-2463.

Wang, Y. M., et al. (2020). "Effect of amendments on soil Cd sorption and trophic transfer of Cd and mineral nutrition along the food chain." Ecotoxicology and Environmental Safety 189.

Response: Agree. Thank you for your very scientific and helpful suggestion. Based on your advice, I have incorporated these two references at appropriate locations in the Introduction section, specifically in lines 58 and 66 of the revised manuscript. Additionally, the references have been updated accordingly. The inclusion of these studies has improved the structure of the manuscript and enhanced the persuasiveness of the introduction. We sincerely appreciate your contribution to strengthening the manuscript.

Once again, we sincerely thank you for your insightful comments and suggestions. Your input has greatly contributed to strengthening the manuscript, and we believe the revisions have enhanced its clarity and scientific rigor. We look forward to your feedback on the revised manuscript.
